# Phase Separation Prevents the Synthesis of VBi_2_Te_4_ by Molecular Beam Epitaxy

**DOI:** 10.3390/nano14010087

**Published:** 2023-12-28

**Authors:** Marieke Altena, Thies Jansen, Martina Tsvetanova, Alexander Brinkman

**Affiliations:** MESA+ Institute for Nanotechnology, University of Twente, 7500 AE Enschede, The Netherlands

**Keywords:** molecular beam epitaxy, VBi_2_Te_4_, magnetic topological insulator, phase separation, crystal growth

## Abstract

Intrinsic magnetic topological insulators (IMTIs) have a non-trivial band topology in combination with magnetic order. This potentially leads to fascinating states of matter, such as quantum anomalous Hall (QAH) insulators and axion insulators. One of the theoretically predicted IMTIs is VBi_2_Te_4_, but experimental evidence of this material is lacking so far. Here, we report on our attempts to synthesise VBi_2_Te_4_ by molecular beam epitaxy (MBE). X-ray diffraction reveals that in the thermodynamic phase space reachable by MBE, there is no region where VBi_2_Te_4_ is stably synthesised. Moreover, scanning transmission electron microscopy shows a clear phase separation to Bi_2_Te_3_ and VTe_2_ instead of the formation of VBi_2_Te_4_. We suggest the phase instability to be due to either the large lattice mismatch between VTe_2_ and Bi_2_Te_3_ or the unfavourable valence state of vanadium.

## 1. Introduction

Over the last decade, the introduction of magnetic order into 3D topological insulators (TIs) has attracted considerable interest. The bandstructure of a TI is characterised by a gapless Dirac cone at the surface, resulting in conducting surface states that are protected by time-reversal symmetry [1]. Magnetism breaks the time-reversal symmetry via exchange interaction and opens a gap in the conducting surface states [1,2,3,4]. This exchange gap can give rise to interesting phases such as the quantum anomalous Hall (QAH) state [2,5,6,7,8,9,10,11] and the axion insulating state [12,13,14,15].

To introduce magnetism into TIs, the following methods are currently used [16]: doping magnetic ions into the TI [6,7,11,17], bringing the TI in proximity with ferromagnetic materials [10,16,18,19,20] and incorporating intrinsic magnetic moments in the crystal structure, which results in an intrinsic magnetic topological insulator (IMTI) [8,9,14,21,22]. All three methods are successful in realising the QAH state; however, with the former two methods the temperatures at which this state arises is very low in the light of applications. It is interesting to compare the temperature at which the QAH effect is observed to the Curie temperature (T${}_{C}$) of the materials. Remarkably, the temperatures for observing the QAH effect are an order of magnitude smaller than T${}_{C}$ [16]. The explanation for this difference in temperature depends on the method used to introduce the magnetism. In the magnetically doped system, the the high level of disorder caused by the random distribution of magnetic dopants may reduce the effective exchange gap [16], form a conducting bulk or create regions without ferromagnetic ordering [11]. In the magnetic proximity system, the sensitivity to the interface between the TI and the magnetic material is the main problem [14].

These challenges are overcome in IMTIs because in these materials the magnetic moment is intrinsically embedded in the unit cell. In 2019, Li et al. [9] theoretically predicted a class of materials acting as IMTIs, called the MBT family (M = transition-metal or rare-earth element, B = Bi or Sb and T = Te or Se). The materials in the MBT family have the same crystal structure, but their behaviour differs depending on the magnetic element (transition-metal or rare-earth element, M) in the MBT structure. The unit cell of the MBT family can be viewed as the unit cell of the well studied family of Bi_2_Te_3_ TIs, with a structural intercalated layer containing a magnetic element. The addition of magnetism within the unit cell results in periodic magnetic layers, which results in a large magnetic exchange gap [9,21]. A representative material of the MBT family is MnBi_2_Te_4_, for which the crystal structure is shown in Figure 1a. Like other materials in the MBT family, it crystallises in the $\bar{R}$3m space group with a rhombohedral structure. Each monolayer has a triangular lattice with ABC stacking along the out-of-plane direction. A monolayer is structured as a septuple layer (SL) with T-B-T-M-T-B-T stacking and a Van der Waals (VdW) gap separates consecutive SLs. The Mn atoms introduce a magnetic moment of 5 μB per atom with an out-of-plane easy axis [9]. The exchange coupling within a single SL (J${}_{\left|\right|}$) is ferromagnetic (FM), while the coupling between consecutive SLs (J${}_{⊥}$) is antiferromagnetic (AFM) [9,23,24,25]. In these VdW materials, the J${}_{\left|\right|}$ is much stronger than J${}_{⊥}$ [23,26].

Another potential member of the MBT family of IMTIs is the theoretically predicted VBi_2_Te_4_ [9]. In contrast to MnBi_2_Te_4_, VBi_2_Te_4_ has a predicted in-plane easy axis (Figure 1b), the V-atoms introduce a magnetic moment of 3 μB per atom [9,26] and a stronger J${}_{⊥}$ is expected in VBi_2_Te_4_ leading to a higher T${}_{C}$[26]. The latter could potentially result in a higher temperature at which topological phases such as QAH can be observed, opening up possibilities for applications. However, to the best of our knowledge, no experimental evidence of VBi_2_Te_4_ has been published so far.

In this work, we report on a structural MBE study to synthesise VBi_2_Te_4_. The crystal structure of the films was analysed by X-ray diffraction (XRD), scanning transmission electron microscopy (STEM) and energy dispersive diffraction (EDX), which are suitable techniques to detect the presence of the SL structure of VBi_2_Te_4_. The surface morphology of the films was characterised by reflective high energy electron diffraction (RHEED) and atomic force microscopy (AFM). The analysis of the crystal structure indicates a phase separation to Bi_2_Te_3_ and VTe_2_ instead of the SL structure of VBi_2_Te_4_. This observation suggests VBi_2_Te_4_ to be unstable in the deposition conditions of MBE.

## 2. Materials and Methods

The deposition of VBi_2_Te_4_ is performed on (0001)-Al_2_O_3_ substrates in an ultrahigh vacuum Octoplus 300 MBE system from Dr. Eberl MBE Komponenten with a base pressure of 5.0 × 10${}^{-11}$ mbar. High-purity (6N) bismuth (Bi) and tellurium (Te) are evaporated from standard Knudsen effusion cells and their fluxes are calibrated by a quartz crystal monitor. The Bi- and Te-flux are kept constant during the depositions at $\phi_{\mathrm{Bi}}$ = 0.0027 Å/s and $\phi_{\mathrm{Te}}$ = 0.072 Å/s. $\phi_{\mathrm{Te}}$ is set to a high flux to prevent Te vacancies. High-purity (5N) vanadium (V) is evaporated from a custom high-temperature Knudsen effusion cell. The flux, $\phi_{V}$, is indicated by the heating temperature of the Knudsen cell and is varied from 1750 ${}^{\circ }$C to 1900 ${}^{\circ }$C. The combination of the V-pocket size and the high evaporation temperature result in a large flux instability measured with the quartz crystal monitor, and therefore the pocket temperature will be kept as a reference for $\phi_{V}$. An estimate for the flux variation in this temperature range is from 0.001 Å/s to 0.0080 Å/s. The substrate temperature T${}_{\mathrm{sub}}$ was kept constant at 150 ${}^{\circ }$C. Before the deposition of VBi_2_Te_4_, a buffer layer of Bi_2_Te_3_ was deposited of ≈1 nm. The samples discussed in this article are deposited using the co-evaporation method, meaning all elemental beams are opened simultaneously during the full deposition. In addition to these results, some attempts were made to use a beam-shuttering method to interrupt the V- and Bi-beams during the deposition. First, Bi and Te are opened to deposit a monolayer of Bi_2_Te_3_. Second, V and Te are opened to deposit a monolayer of VTe on top of the Bi_2_Te_3_ layer. Third, an annealing step is applied during which the VTe layer should diffuse into Bi_2_Te_3_ to form the SL of VBi_2_Te_4_. These three steps were repeated to form a multilayered VBi_2_Te_4_ film. This method was previously used to successfully deposit MnBi_2_Te_4_ by MBE [27], but for VBi_2_Te_4_ the beam-shuttered method resulted in the same observations discussed here for the co-evaporation method showing a phase separation to VTe_2_ and Bi_2_Te_3_. Right after deposition, a RHEED image of the diffraction pattern is taken. From the RHEED image the in-plane lattice constant can be deduced by comparing the diffraction pattern of the film to a known substrate.

The crystal structure of the films is measured with XRD, STEM and EDX. The XRD measurements are performed with a Bruker D8 Discover system (Bruker, Billerica, MA, USA) with a two-dimensional Eiger2 500K detector and a two-bounce channel-cut germanium monochromator. Symmetric 2θ-ω scans were performed along the surface normal direction. The STEM measurements are made with a Thermo Scientific Spectra 300 STEM (Thermo Fisher Scientific, Waltham, MA, USA) with an electron beam voltage of 300 kV and a high-angle annular dark-field (HAADF) detector.

## 3. Results

The crystal structure of the films is analysed with STEM, EDX and XRD. STEM is performed on a sample deposited with $\phi_{V}$ = 1800 ${}^{\circ }$C (Figure 2a). The image shows the V-Bi-Te film and the Al_2_O_3_ substrate. These STEM results clearly indicate two regions by looking at the contrast. These variations are caused by the Z-contrast related to the atomic weight of the present elements. For a V-Bi-Te sample, the atomic weights are arranged as $m_{\mathrm{Bi}}$ > $m_{\mathrm{Te}}$ > $m_{V}$. Therefore, the bright areas in Figure 2a are Bi-rich regions. These results clearly indicate a phase separation between a Bi-compound and a non-Bi compound. EDX (Figure 2b) shows a clear separation between a Bi/Te region and a V/Te region. Figure 2c shows a detailed STEM scan of the sample. The atoms in the bright areas are structured as a QL separated by a VdW-gap. This structure is consistent with Bi_2_Te_3_ (Figure 1c). In the darker area the bright atoms form the typical Te octahedra of VTe_2_ which are separated by a VdW gap as shown in Figure 1d. The in-plane lattice constant *a* is related to the distance between the atoms in the x-direction, d*x*. Figure 2d shows the distribution of d*x* as extracted from Figure 2c. This distribution indicates two clearly separated regions. The in-plane lattice constants related to these two regions are calculated from the d*x* value with maximum intensity as *a* = 2*d*x for both Bi_2_Te_3_ and VTe_2_. This calculation results in the lattice constants $a_{1}$ = 3.82 Å, corresponding to VTe_2_, and $a_{2}$ = 4.65 Å, corresponding to Bi_2_Te_3_.

Figure 2e presents the 2θ-ω scans of films deposited with different $\phi_{V}$. The peaks in the 2θ-ω scans can be identified as the (00l)-Bi_2_Te_3_ and (00l)-VTe_2_ peaks. The dotted arrow at the (006)-Bi_2_Te_3_ peak indicates the dominance of Bi_2_Te_3_ at low $\phi_{V}$, but the intensity of this phase decreases as $\phi_{V}$ increases. The dashed arrow at the (001)-VTe_2_ peak indicates the dominance of VTe_2_ at high $\phi_{V}$, but this phase disappears as $\phi_{V}$ decreases. VBi${}_{2}$Te${}_{4}$ is absent in all 2θ-ω scans.

The surface of the films is analysed with RHEED and AFM. Figure 3a presents the in situ RHEED pattern of a film deposited with $\phi_{V}$ = 1750 ${}^{\circ }$C. The RHEED pattern consists of a double streak pattern as indicated by the blue and white arrows. This doubled pattern indicates the presence of two separate crystal phases at the surface. The in-plane lattice constants related to these streaks are $a_{1}$ = 4.31 Å (white arrows) and $a_{2}$ = 3.59 Å (blue arrows). These values correspond well with the lattice constants of Bi_2_Te_3_ and VTe_2_, respectively.

The surface morphology is measured by AFM. Figure 3b shows the height distribution as measured with AFM for films deposited with different $\phi_{V}$. The insets show the surface morphology of the films with $\phi_{V}$ = 1750 ${}^{\circ }$C and $\phi_{V}$ = 1900 ${}^{\circ }$C. At low $\phi_{V}$, the morphology shows strong island formation and the triangular crystals typically observed for Bi_2_Te_3_. The results at high $\phi_{V}$ show a relative flat film without any sharp crystals. The height distributions indicate a strong influence of the $\phi_{V}$ on the distribution spread. With an increasing $\phi_{V}$, the height variation becomes smaller, indicating a flatter surface.

## 4. Discussion

VBi_2_Te_4_ is a SL structure requiring the embedding of VTe within the QL structure of Bi_2_Te_3_. According to our results, the formation of VBi_2_Te_4_ is unstable with respect to phase separated Bi_2_Te_3_ and VTe_2_ within the thermodynamic conditions of the MBE. The instability of VBi_2_Te_4_ can have various causes.

First, a large in-plane lattice mismatch, $\Delta_{a}$, between VTe_2_ and Bi_2_Te_3_ might prohibit the formation of VBi_2_Te_4_ [28]. Our STEM results indicate $\Delta_{a}$ = 0.83 Å between the two phases. The theoretically predicted lattice constant of VBi_2_Te_4_, *a* = 4.37 Å, is close to the lattice constant of Bi_2_Te_3_, *a* = 4.65 Å. Therefore, the VTe_2_ lattice has to overcome $\Delta_{a}$ to form VBi_2_Te_4_. Table 1 gives an overview of different materials structured as a SL with the relevant lattice constants and whether the material was successfully observed in experiments. The intercalated layer is presented as XTe or XTe${}_{2}$, depending on the experimental stability of the phases. The experimentally successful materials match $\Delta_{a}$ < 0.6 Å, while the experimentally unsuccessful materials match $\Delta_{a}$ > 0.5 Å. This difference can indicate a limit to the maximum allowed $\Delta_{a}$ of 0.5 Å to 0.6 Å between the SL material and the intercalated layer, possibly explaining the phase separation in VBi_2_Te_4_. However, this observation does not match with the stability of PbBi_2_Te_4_ and SnBi_2_Te_4_ [29]. Therefore, the lattice mismatch between the intercalated layer and Bi_2_Te_3_ does not completely explain the instability of the SL structure in general, and another factor should be considered.

Second, the elemental valence states in the intercalated layer might prohibit the formation of the SL. In the SL, the preferred valence states are M${}^{\left(+2\right)}$Bi_2_${}^{\left(+3\right)}$Te_4_${}^{\left(-2\right)}$ (M = transition metal or rare-earth element), which matches well with the valence states of an intercalated layer structure of M${}^{\left(+2\right)}$Te${}^{\left(-2\right)}$ [30]. However, when the stable compound of the intercalated layer is structured as M${}^{\left(+4\right)}$Te_2_${}^{\left(-2\right)}$ the valence states of the intercalated layer and the SL do not match. A mismatch between the preferred valence state of the intercalated layer and the SL indicates the instability of the SL. Table 1 reflects this instability, showing that every experimentally observed intercalated layer bulk compound with a valence structure of M${}^{\left(+2\right)}$Te${}^{\left(-2\right)}$ also has a stable SL counterpart, but a valence structure of M${}^{\left(+4\right)}$Te_2_${}^{\left(-2\right)}$ does not. This is in agreement with our study on VBi_2_Te_4_, because VTe_2_ is thermodynamically more stable than VTe [31].

Furthermore, ref. [32] studied the preferred valence states of V, Cr, Mn and Fe in Bi_2_Te_3_. In Te-rich conditions, only V${}^{3+}$ and Cr${}^{3+}$ can substitute neutrally for Bi${}^{+3}$ atoms in Bi_2_Te_3_. In contrast, Mn and Fe mostly form Mn${}^{2+}$ and Fe${}^{2+}$, which create energetically unfavourable states when mixed with Bi${}^{3+}$ [9,32]. This additionally shows the unfavourable V${}^{2+}$ valence state. Therefore, Mn and Fe can more easily form a neutral SL structure with respect to V and Cr.

**Table 1 nanomaterials-14-00087-t001:** Overview of materials with a unit cell structured as a SL. The table presents whether the material is successfully synthesised experimentally, the lattice constants found in the literature for these materials (either experimental or theoretical values), the intercalated layer with the corresponding lattice constant and the lattice mismatch between the SL and the intercalated layer. The intercalated layer is presented as XTe or XTe_2_, depending on the experimental stability of the phases.

Material	Experimentally Observed?	a${}_{\mathrm{SL}}$ [Å]	Intercalated Layer	a${}_{\mathrm{int}.}$ [Å]	Lattice Structure	$\Delta_{a}$ [Å]
VBi_2_Te_4_	No	4.34–4.37 [9,26,33,34,35]	VTe_2_	3.59	Hexagonal, P$\bar{3}m$1	0.75–0.78
MnBi_2_Te_4_	Yes [8,21,22,23,27,36]	4.33 [36]	MnTe	4.13 [37]	Hexagonal, P6_3_/mmc	0.20
FeBi_2_Te_4_	Yes [38]	4.39 [9,38]	FeTe	3.83 [38,39]	Tetragonal, P4/nmm	0.56
			FeTe_2_	3.77 [40]	Hexagonal, P$\bar{3}m$1	0.63
EuBi_2_Te_4_	No	4.50 [9,33]	EuTe	6.60 [41,42]	Cubic, Fm$\bar{3}m$	2.10
			EuTe_2_	6.97 [43]	Tetragonal, I4/mcm	2.47
NiBi_2_Te_4_	Yes *	4.30 [9,33]	NiTe_2_	3.86 [44]	Hexagonal, P$\bar{3}m$1	0.44
CrBi_2_Te_4_	No	4.32 [45]	CrTe_2_	3.79 [46]	Hexagonal, P$\bar{3}m$1	0.53
TiBi_2_Te_4_	No	4.39 [9]	TiTe_2_	3.78 [47]	Hexagonal, P$\bar{3}m$1	0.61
PbBi_2_Te_4_	Yes [48,49]	4.44 [49]	PbTe	6.46 [50]	Cubic, Fm$\bar{3}m$	2.02
SnBi_2_Te_4_	Yes [51,52,53]	4.40 [51,53]	SnTe	6.32 [50]	Cubic, Fm$\bar{3}m$	1.92
GeBi_2_Te_4_	Yes [54,55]	4.33 [54,55]	GeTe	4.16 [50,56]	Rhombohedral, R$\bar{3}m$	0.17

* Not observed as multilayered/bulk material. Ref. [57] observed the SL structure as an intercalated layer between Bi_2_Te_3_ and Ni-doped Bi_2_Te_3_.

In conclusion, the influence of the $\phi_{V}$ during MBE depositions was investigated on the synthesis of the VBi_2_Te_4_ phase. The resulting films do not show any indication of VBi_2_Te_4_ but rather a phase separation into Bi_2_Te_3_ and VTe_2_. These results show VBi_2_Te_4_ is unstable within the deposition conditions of MBE.

## Figures and Tables

**Figure 1 nanomaterials-14-00087-f001:**
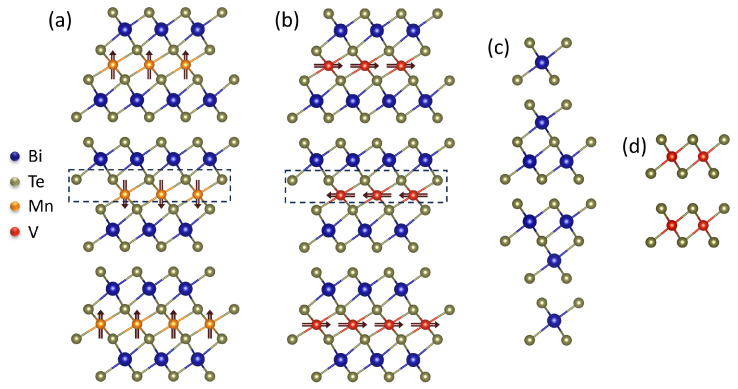
(**a**) MnBi_2_Te_4_ and (**b**) VBi_2_Te_4_ have a unit cell structured as SLs separated by a VdW gap. The dashed boxes indicate the relative intercalated layers of MnTe and VTe in Bi_2_Te_3_. J${}_{\left|\right|}$ is FM with either an (**a**) out-of-plane or (**b**) in-plane easy axis. J${}_{⊥}$ is AFM. (**c**) Bi_2_Te_3_ structured in QLs separated by a VdW gap. (**d**) VTe${}_{2}$.

**Figure 2 nanomaterials-14-00087-f002:**
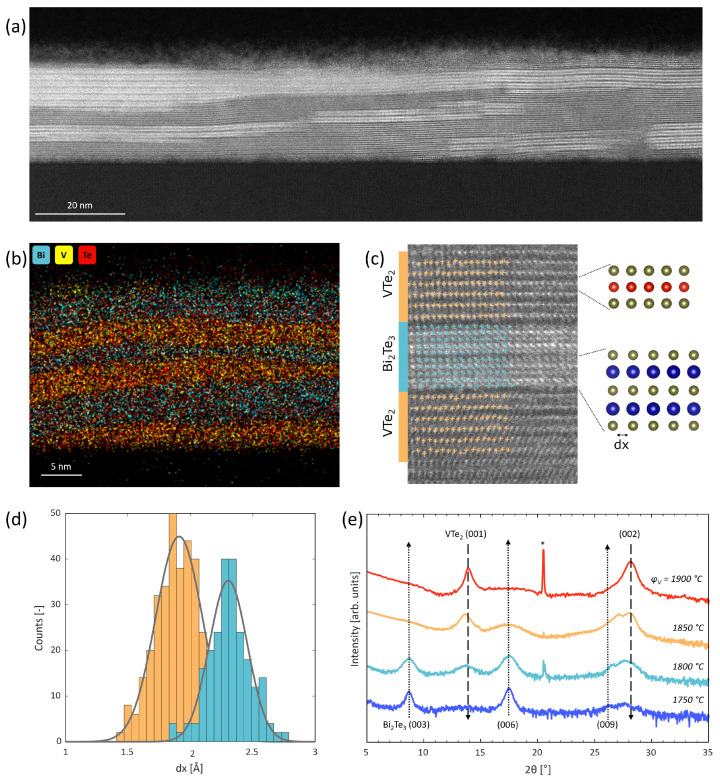
(**a**) STEM image of a V-Bi-Te sample. The image is taken with a HAADF detector at 300 keV. A clear phase separation between bright and dark areas can be observed. (**b**) EDX scan of the V-Bi-Te sample. A strong separation between V-regions and Bi-regions can be observed. (**c**) STEM image of a smaller region on a V-Bi-Te sample. The bright areas (blue) show the QL structure of Bi_2_Te_3_ and the dark areas (orange) the VTe${}_{2}$ structure. (**d**) Histogram of the atomic distance in the x-direction. (**e**) 2θ-ω scans indicating (00l)-Bi_2_Te_3_ being dominant at low $\phi_{V}$, while (00l)-VTe${}_{2}$ is dominant at high $\phi_{V}$. The arrows indicate the disappearance of the respective phases as a function of $\phi_{V}$. * indicates the Al_2_O${}_{3}$ substrate peak.

**Figure 3 nanomaterials-14-00087-f003:**
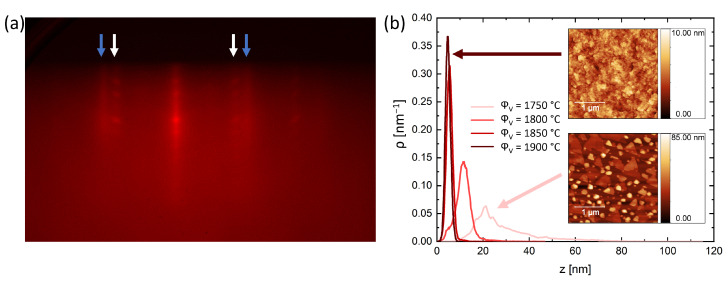
(**a**) RHEED pattern for $\phi_{V}$ = 1750 ${}^{\circ }$C showing a double streak pattern related to the phases Bi${}_{2}$Te${}_{3}$ with $a_{1}=$ 4.31 Å and VTe${}_{2}$ with $a_{2}=$ 3.59 Å, indicated by the blue and white arrows, respectively. (**b**) Height distribution at the surface as a function of the $\phi_{V}$. The insets show the surface morphology of the samples with $\phi_{V}$ = 1750 ${}^{\circ }$C and $\phi_{V}$ = 1900 ${}^{\circ }$C.

## Data Availability

The data presented in this study are openly available in 4TU.ResearchData at 10.4121/95ee6f7b-3cf7-4c01-a461-860e6f9485bf.

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
