# Peer review of "Phase Separation Prevents the Synthesis of VBi2Te4 by Molecular Beam Epitaxy"

_nanomaterials, 2023, doi:10.3390/nano14010087_

Round 1
Reviewer 1 Report
Comments and Suggestions for Authors
In this work, the authors reported a molecular beam epitaxy (MBE) study aiming to synthesize VBi2Te4, a septuple layer (SL) structure. The crystal structure analysis using various techniques, such as X-ray diffraction (XRD), scanning transmission electron microscopy (STEM), energy dispersive diffraction (EDX), reflective high energy electron diffraction (RHEED), and atomic force microscopy (AFM), reveals a phase separation to Bi2Te3 and VTe2 rather than the desired SL structure of VBi2Te4. In my opinion, this work can be considered for publication in Nanomaterials with the following revisions: 1. The discussion suggests a limit (∆a < 0.6 Å) for successful synthesis based on lattice mismatch. How does this threshold compare with other successful SL materials, and are there exceptions that need consideration? Since, the author has mentioned that this observation does not match with the stability of PbBi2Te4 and SnBi2Te4. 2. To what extent can the observed instability be attributed to the thermodynamic stability of VTe2 compared to VTe, and are there other factors influencing valence states that need exploration? 3. Are there alternative explanations for the observed phase separation, considering the complexity of factors such as valence states and lattice mismatch? 4. The AFM results suggest island formation at low V flux. How does this morphology correlate with the proposed phase separation, and what implications might it have for film quality? 5. The authors are suggested to compare and discussion some recent works (i.e. Ultrafast Science, 3, 0006, 2023; Advanced Photonics, 2020, 2(2): 024001).
Reviewer 2 Report
Comments and Suggestions for Authors
VBi2Te4 is one of the theoretically predicted intrinsic magnetic topological insulators (IMTIs). However, it is still lack of experimental evidence to prove its existence. In this paper, the authors report on their attempts to synthesise VBi2Te4 by MBE growth. Instead of obtaining positive results, they reveal the unstably synthesis of VBi2Te4 by MBE and present the possible causes leading to the instability of VBi2Te4. Although the new substance VBi2Te4 predicted by theory has not been obtained by MBE, this paper has important reference value for people to explore VBi2Te4 by other methods. The whole manuscript is well organized and the research design is appropriate. The results are also clearly presented for the readers. I recommend its publication as it is.
Comments on the Quality of English Language
Please correct misprints.
Author Response
We would like to thank the reviewer for his/her kind words and careful evaluation of our manuscript.
Reviewer 3 Report
Comments and Suggestions for Authors
The article fits perfectly into the theme of the magazine. The literature review is factual and relates directly to the topic of the article. The experiment raises no objections. Moreover, what is important in my opinion, the authors do not include unnecessary figures and do not include unnecessary descriptions. I believe that the publication is very well thought out and can be published without changes.
Author Response

(The authors gave the same response as above.)

Reviewer 4 Report
Comments and Suggestions for Authors
The manuscript presents an investigation of growth of VBi2Te4 crystal films by molecular beam epitaxy. This crystal is theoretically predicted to be a magnetic topological insulator which is an interesting state of matter. The presentation is well prepared with clear figures and explanations. However, some corrections are needed.
Comments
1. Did you measure the QAH effect in your samples? You have stated it in Abstract, and thus need to discuss. Otherwise, remove it from Abstract.
2. Details of the MBE growth are unclear. Your statement in conclusion “These results show VBi2Te4 is unstable within the deposition conditions of MBE.” is confusing. What deposition conditions? Did you use beam shutters/ interruptions? Can you give the vanadium flux value instead of temperature? Please clarify.
3. Did you observe a formation of superlattice with a unit (VTe/Bi2Te3) ? It would be interesting. Fig.2 (b,c) seems indicate on such possibility. This could be a positive outcome of your work instead of the negative.
4. Table 1, what is the meaning of “intercalated layer” here? Is it the layer between neighbor QLs of Bi2Te3? Fig.1 does not show it. Please specify.
Comments on the Quality of English LanguagePlease correct misprints.
